# Analysis and Comparison of Two Artificial Intelligence Diabetic Retinopathy Screening Algorithms in a Pilot Study: IDx-DR and Retinalyze

**DOI:** 10.3390/jcm10112352

**Published:** 2021-05-27

**Authors:** Andrzej Grzybowski, Piotr Brona

**Affiliations:** 1Department of Ophthalmology, University of Warmia and Mazury, Żołnierska 18, 10-561 Olsztyn, Poland; 2Institute for Research in Ophthalmology, Foundation for Ophthalmology Development, 60-836 Poznan, Poland; 3Department of Ophthalmology, Poznan City Hospital, Szwajcarska 3, 60-285 Poznan, Poland; piotrbrona@gmail.com

**Keywords:** diabetic retinopathy, diabetic eye disease, artificial intelligence, machine learning, deep learning, diabetic retinopathy screening, ophthalmology, diabetology, public health

## Abstract

Background: The prevalence of diabetic retinopathy (DR) is expected to increase. This will put an increasing strain on health care resources. Recently, artificial intelligence-based, autonomous DR screening systems have been developed. A direct comparison between different systems is often difficult and only two such comparisons have been published so far. As different screening solutions are now available commercially, with more in the pipeline, choosing a system is not a simple matter. Based on the images gathered in a local DR screening program we performed a retrospective comparison of IDx-DR and Retinalyze. Methods: We chose a non-representative sample of all referable DR positive screening subjects (*n* = 60) and a random selection of DR negative patient images (*n* = 110). Only subjects with four good quality, 45-degree field of view images, a macula-centered and disc-centered image from both eyes were chosen for comparison. The images were captured by a Topcon NW-400 fundus camera, without mydriasis. The images were previously graded by a single ophthalmologist. For the purpose of this comparison, we assumed two screening strategies for Retinalyze—where either one or two out of the four images needed to be marked positive by the system for an overall positive result at the patient level. Results: Percentage agreement with a single reader in DR positive and DR negative cases respectively was: 93.3%, 95.5% for IDx-DR; 89.7% and 71.8% for Retinalyze strategy 1; 74.1% and 93.6% for Retinalyze under strategy 2. Conclusions: Both systems were able to analyse the vast majority of images. Both systems were easy to set up and use. There were several limitations to the current pilot study, concerning sample choice and the reference grading that need to be addressed before attempting a more robust future study.

## 1. Introduction

Diabetes is a global epidemic with the number of people affected on a constant rise. Prevalence of diabetic retinopathy (DR), a complication of diabetes, is expected to increase accordingly [1]. This puts an increasing strain on health care resources. Currently only a few high-income countries were able to establish a nationwide DR screening based on tele-medicine, including United Kingdom and Singapore. These programs were shown to be effective, decreasing the number of DR-related blindness in these countries; however, they are very expensive, which makes them unavailable for many other countries [2]. Conventional human-based grading is not only expensive in terms of monetary cost, but also requires specially trained graders or ophthalmologists, both of which are often in short supply. An AI-based screening solution could grade thousands of images every day, in multiple places at once, with relatively little human capital cost. Recent advances in computer science and machine learning made artificial intelligence (AI) based DR screening a possibility, with a number of software already commercially available. 

Retinalyze (RetinaLyze System A/S, Copenhagen, Denmark) is a cloud-based, fundus image analysis software, offering automated screening for DR, age-related macular degeneration, and glaucoma. It is one of the earliest introduced automated DR screening software. It was initially described over 15 years ago, showing its lesion-detection based screening on images taken on 35 mm film [3,4]. Since its initial introduction it has undergone updates to bring it in line with todays practice. The system accepts fundus images sent over a browser-based interface. It is unclear whether deep-learning aspects are used in the iteration of Retinalyze that this study was used on. The results are displayed in terms of the number of abnormalities detected in the image. Retinalyze is available in Europe. It is not currently approved for use in North America, other than Mexico. 

IDx-DR (Digital Diagnostics, Coralville, IA, USA) is an autonomous AI system for the real time, point of care diagnosis of diabetic retinopathy and diabetic macular oedema. IDx-DR started as a project at the University of Iowa, under the name of the Iowa Detection Program (IDP). Initially it was composed of expert-designed algorithms with deep-learning components being introduced much later. It is autonomous, which means that no human oversight of the clinical decision is needed. It includes an operator assistive AI to help operators without any imaging proficiency take high quality images of the retina, assisting where needed in retaking images which are of insufficient quality, the wrong area, or out of focus. It is the first autonomous AI system with U.S. Food and Drug Administration (FDA) clearance for marketing in USA. The study that paved the way for IDx-DR to be approved was conducted in a primary care setting with operators without any previous experience in retinal imaging, and compared to a patient outcome proxy, compared to professional four wide-field stereoscopic fundus images and OCT data obtained by certified retinal photographers. The photographs were read by an independent reading center that created the patient outcome reference standard used for all DR management trials, the ETDRS [5]. It is designed to diagnose Early Treatment Diabetic Retinopathy Study (ETDRS) level 35 or more DR or clinically significant or center involved diabetic macular edema (DME). Very recently, an additional DR analysis software—Eyeart (Eyenuk, Woodland Hills, CA, USA) also received FDA approval. In Europe, IDx-DR is certified as a class IIa medical device.

A number of artificial intelligence solutions for use in DR screening programs, some authorized for autonomous use, are currently available with more in the pipeline [6]. While many of those have research papers to show their effectiveness [5,7,8,9,10], a direct real-life comparison is often very difficult due to a number of factors. We attempted a pilot DR opportunistic screening programme. The screening comprised of non-mydriatic fundus imaging, done in Poznan, Poland [11]. We decided to analyse two of the leading, commercially available screening systems—IDx-DR and Retinalyze.

IDx-DR is registered in the US for use with Topcon Nw-400 non-mydriatic camera, and the operator assistance is required to be used to achieve the reported diagnostic accuracy and diagnosability, which was used for taking all of the images in this study. IDx-DR requires all four images to perform the screening. Retinalyze is a web-based service that analyses each image separately and labels each image with regards to whether retinal abnormalities have been detected or not. It also provides an annotated version of the image with the suspected DR changes highlighted. The results are per-image and each image is screened separately.

## 2. Methodology 

Patients who visited a community-based, secondary care, diabetic clinic in Poznan, Poland for their routine appointments were offered DR screening based on one disc- and one macula-centred image per eye using a non-mydriatic camera—TRC-NW400, Topcon. The clinic did not previously conduct DR screening and existing staff were purpose trained in taking the images. Diabetics older than 18 years old, who were not known to have DR, were screened on an opt-in basis, after signing a written consent form. Both eyes were imaged with two images taken per eye—a macula-centered and a disc-centered photo for a total of four photos per patient. All images were taken without mydriasis. There were no further exclusion criteria.

Due to constraints regarding the volume of patients we could screen using the two algorithms, the group chosen for comparison consisted of all patients with referable DR (rDR)–-60 patients, 240 images total, and a randomly chosen group of DR negative patients—110 patients, 440 images total. Only patients with all four retinal images of sufficient quality for human grading, that is images for which patient’s DR level could be distinguished, were chosen. As a result, the AI systems’ image quality feedback systems were not evaluated. This excluded approximately 25% of patient encounters. The images were graded by a single ophthalmologist with basic experience in DR screening, and a final result of rDR positive or negative on a per patient basis was set. Referrable DR was defined as more than mild DR according to the International Clinical Diabetec Retinopathy (ICDR) scale. The ophthalmologist reader graded the images before reviewing the output of the two systems. 

In order to compare the two systems, we devised two screening strategies using Retinalyze, setting the threshold for overall rDR positive status as at least one (Strategy 1) or two (Strategy 2) positive images per patient. This was devised for the purpose of this study as a means of a rudimentary comparison between the two systems and is not a screening strategy sanctioned by any of the compared algorithms. 

European conformity certification allows for the use of medical devices inside of the European Economic Area. We have asked the two companies for documents regarding their certification within Europe. The study is based on retrospective analysis of images used for clinical purposes, and patients consented that their images will be used in scientific analyses. The ethics approval in such studies in Poland is not required. 

For the statistical analysis, binomial confidence limits were calculated and displayed as the Clopper-Pearson confidence intervals. The points on the ROC curve were generated using each possible outcome of the diagnostic test as a classification cut-point and computing the corresponding “Sensitivity” and “1—Specificity”. The area under the resulting ROC curve (AUC) was computed using the trapezoidal rule. The default standard error for the area under the ROC curve was calculated using the algorithm proposed by DeLong, DeLong, and Clarke-Pearson. Youden’s J statistics (index) was computed based on the formula: J = Sensitivity + Specificity – 1.

## 3. Results

For the rDR positive group, consisting of 60 patients, IDx-DR was able to analyse all 60 patients while Retinalyze was able to analyse all images for 58 patients and gave a result of insufficient quality on 1 out of 4 images for 1 patient. For this group, overall IDx-DR agreement was 93.33% (95% CI: 83.80–98.15%) compared to a single reader and Retinalyze was 89.66% (95% CI: 78.83–96.11%), (at least 1) or 74.14% (95% CI: 60.96–84.74%) (at least 2 positive). IDx-DR marked 4 out of the 60 patients as negative while Retinalyze Strategy 1 marked 6 such patients as negative. A breakdown of the software’s results for rDR positive group is shown in Table 1.

For the DR absent group, both systems were able to analyse all 480 images, with IDx-DR achieving a 95.45% (95% CI: 89.71–98.51%) agreement compared to a single reader, Retinalyze achieving an agreement of 71.82% (95% CI: 62.44–79.98%) for Strategy 1 or 93.64% (95% CI: 87.33–97.40%) for Strategy 2. 

A breakdown of the software’s results for rDR negative group is shown in Table 2.

The two strategies used in this study for designating a final patient level result from individual image results of Retinalyze giving different results with Strategy 2 (two positive images for an overall positive result), having higher specificity at the cost of lower sensitivity. The reverse occurred with Strategy 1, with high sensitivity and better negative predictive value at 92.94%, versus 87.29% for the second strategy. Results of the overall statistical analysis are shown in Table 3.

## 4. Discussion

A direct comparison between AI DR screening software is hampered by not only significant differences in the way systems structure and present results, but also because they are tuned for detecting different levels of DR. This difficulty is further compounded by issues surrounding human grading and its reliability in establishing the ground truth. As a result, some of the more robust trials evaluating a given AI DR system opted for the use of 7-field images, stereoscopic images and even OCT scans to aid human graders [5,9].

Previously, after screening 400 patients, we have found the IDx-DR sensitivity and specificity, compared to a single reader, to be 94% and 95%, although we have taken a pragmatic approach and decided to count other retinal pathology, detected as referable DR, as an accurate result [11]. This has shown that the AI algorithm had good overall accuracy in our local population. Therefore, we expected similar or slightly lower accuracy results for IDx-DR for this study, which was confirmed with IDx-DR achieving an agreement level of about 93–94% for both DR positive and negative patients.

One of the common objections regarding deep learning based DR screening is the black box effect, meaning we are presented with a result but have no way of knowing how a system reached the given conclusion [12]. This is particularly important in cases where a system outputs a result other than expected. IDx-DR uses an explainable AI, by using multiple algorithms that separately look specifically for different types of lesions, much as a clinician would. Although the individual calculations and their results, processed by IDx-DR, are not available to the end-user, these can be accessed by the manufacturer and oversight or governing bodies such as the FDA. The annotated image approach used by Retinalyze shows regions the system deems suspicious for DR. This is helpful in the further care of a given patient with a positive screening result. A human grader or clinician can look at the annotated software result and focus the examination to the relevant area. Nevertheless, any patients presenting with a positive result require a human grader verification and/or an ophthalmological examination regardless of having an annotated result or not.

There is a significant lack of comparison studies using modern, current-generation AI solutions, which makes choosing any particular screening solution difficult, with only two other comparison studies currently available. The first study, that we are aware of, to directly compare different named autonomous systems for DR detection was published in 2016 by Tufail and colleagues [13]. This study focused on assessing autonomous DR screening software’s viability for replacing human graders as part of the UK national screening programme. Both accuracy and cost-effectiveness were considered. Eyeart and RetmarkerDR (RETMARKER, S.A., Taveiro, Portugal) were shown to have 94.7% and 73% sensitivities for any DR, compared to the 94%, 87%, and 73% sensitivity results in our study [13]. In Tufail’s comparison, the systems suffered from low specificity, with showing results of 20% for Eyeart and 53% for RetmarkerDR (RETMARKER, S.A., Taveiro, Portugal) [13]. It is important to highlight that two different software packages were compared in this study. We found substantial higher agreement with a single reader, regarding DR negative patients of 93% for IDx-DR and 73%, 90% for Retinalyze depending on strategy used. A contributing factor to this discrepancy could be the greater robustness of Tufail’s study as compared to the single reader grading used here. In Tufail’s study both multiple experienced DR graders were used, as per the national DR screening protocol, and an external reading centre was employed to adjudicate results. Additionally, a larger volume of images were used—over 100,000 fundus images from over 20,000 patients. Another difference between Tufail’s study and this paper is related to the rapid development and later widespread adoption of machine-learning in DR screening solutions. Although Tufail’s study was published in 2016, the trial started in 2013 [13]. The introduction of machine-learning techniques, even for already established lesion-based screening systems, allowed for greater accuracy [5,10]. Abramoff and colleagues compared two versions of the same software, before and after introducing deep-learning based analysis into the already established expert-designed algorithm. Comparing those on the Messidor-2, a publicly available dataset has shown significant improvement in accuracy after deep-learning measures were implemented [14].

The second prospective, multi-centre comparison study was recently published by Lee and colleagues [15]. The study compared 7 algorithms from 5 different companies against screening encounters at two different hospitals, for a total of 23,724 patient encounters. The algorithm names, in the aforementioned study’s results, have been anonymised to encourage participation [15]. Neither IDx-DR nor Retinalyze participated in this comparison. The study used a robust grading protocol with adjudication and set the final grade as either referable DR or no referrable DR. The performance of different AI algorithms was also compared to a single teleretinal grader that originally graded the given patient, somewhat similarly to the single reader used in this study. The accuracy results varied significantly between the algorithms with only 3 out of 7 achieving comparable sensitivity and one achieving comparable specificity to the original teleretinal graders [15]. 

In Europe, medical devices including software are subject to ‘The Medical Devices Regulation’ (2017/745/EU). This legislation requires medical software manufacturers to risk-stratify their products and follow one of the certification pathways according to the risk level. Class I, or low risk, devices can be self-certified by the manufacturer and require no external oversight or governing body control. However, according to the aforementioned legislation, software “intended to provide information, which is used to take decisions with diagnosis”, necessitates a certification level of at least IIa. As such, according to the Conformité Européenne (CE) marking, IDx-DR, possessing a class IIa certification, is compliant and can be used without a clinician verifying its outputs.

Unlike the way medical devices are regulated in the USA, with FDA being the central supervisory and certifying body, there is no such central authority in the EU. Certification is done through accredited third party, private entities called “Notified Bodies” [16]. There is no publicly available register of EU approved medical devices, and information sent to Notified Bodies and regulators is often confidential [16]. There are plans to alleviate this, at least partly, with the introduction of Eudamed database in 2022 [16]. Eudamed is supposed to contain registration of certificates of conformity, summary of safety, and clinical performance accessible by the public. Unfortunately, until that is introduced, we are reliant on the goodwill and transparency of the individual companies in obtaining such information.

To our knowledge, Retinalyze possessed a legacy Class I certification only (obtained before the current legislature in regards to diagnostic software was put into place), and unless that was updated to at least class IIa, cannot be used for diagnosing patients without direct clinician oversight. This would make it unsuited for any DR-screening program, as it doubles the cost of the grading. Unfortunately, the company behind Retinalyze refused us the most recent update on Retinalyze’s CE certification.

The two pieces of software described here operate differently. IDx-DR requires 4 fundus images and produces an overall per patient result, while Retinalyze analyses each image separately and gives a per-image result. IDx-DR is certified to be used with the Topcon NW-400 fundus camera, although it does work with other image sources, while Retinalyze does not specify any parameters for the submitted images. IDx-DR is designed explicitly for a cut-off point of “more than mild DR” while Retinalyze displays its results in terms of retinal abnormalities detected or not detected with those regions highlighted. 

### Limitations

The results of this study are hampered by several limitations, that need to be addressed in designing a future, more robust comparison. Firstly, the sample selected was not representative of the diabetic population. We were forced to constrain this study to a certain number of patients, and DR positive cases were largely overrepresented in this study, comparing to what we have seen in the screening project as a whole, leading to a selection bias.

Sensitivity, specificity of the single grading ophthalmologist is unknown and was not compared to other readers or a reading centre such as the Wisconsin Reading Center, or ETDRS. The grader had access to the patient images only, with no other clinical or examination data available. Patient outcome-based proxy for truth was not available, so that the single reader had access to the same information as the AI systems had, and not to OCT or widefield stereo of a larger retinal area—thus overestimating performance.

The software to be compared was chosen based on its availability to us. More automated DR diagnostic software could be sought out, for a broader comparison. In terms of the dataset used, although it was taken from a real-life screening scenario, low image quality encounters were purposely excluded, therefore not examining the software’s ability to deal with lower quality images. The effect of ungradable images was not studied, but is required for establishing diagnostic test accuracy in a real-life application. 

Because of the relatively low rate of DR positive results in our screening programme, to achieve a meaningful diagnostic test accuracy measurement, the overall sample size would need to be much bigger. Unfortunately, due to the constraints in the number of images we could test, this could not have been done for this pilot study. 

The standard to which the AI systems were held was created by a single grader. Grading of DR can be a fairly complex process, with a significant amount of intra and intergrader variability and varying grader accuracy [17]. Employing two or three graders, in addition to an adjudication process, would strengthen the gold standard considerably but requires a significantly more resources. The current benchmark for setting up a reference standard in the field of automated retinal image analysis is multiple human graders, preferably with specialty adjudication. This is done to reduce the influence of grader bias, mistakes, and intra-grader variability on the final result.

Comparing those two AI DR systems is additionally challenging as they function and analyse the patient encounter at different levels. IDx-DR generates an overall per-patient result, while Retinalyze presents a per-image result. Comparing the two systems directly is not possible and therefore the comparison depends on making suppositions regarding their implementation into a screening setting. The screening strategy that we employed in order to compare the two systems, with the four image per patient database we had, relied on submitting all four images to both algorithms. This is the ideal use case for IDx-DR, which requires four good quality images in order to produce a result. However, Retinalyze can be used with any number of images from a given patient, with a result for each. This may be advantageous in situations where obtaining four good quality images is difficult due to media opacities or other issues. Should a different screening procedure be considered, for example using a single macula-centered image per eye, one might expect significantly different results. 

## 5. Conclusions

Compared to a single reader, both systems were able to achieve good accuracy results, with very few false-negative results. Both systems were able to analyse the vast majority of images, keeping in mind the dataset was curated for quality. Currently available DR diagnostic systems differ in key operational parameters, such as analysing single images with a result for each or outputting a single grade for all patient’s images. These disparities confound attempts at comparing those systems directly. A rigid protocol for grading, preferably with adjudication and/or external grading involvement would be crucial in establishing a reliable ground truth for comparison. An actionable, direct comparison of AI DR systems is difficult and would require addressing the limitations of this study, particularly in terms of a larger sample size and a robust protocol for establishing ground truth.

## Figures and Tables

**Table 1 jcm-10-02352-t001:** Screening software results for the DR present group (numbers represent patients, each with 2 images per eye).

	IDX-DR	Retinalyze Strategy 1	Retinalyze Strategy 2
Marked as DR detected	56	53	44
Marked as no DR	4	6	15
Marked as insufficient quality	0	1	1

DR—diabetic retinopathy.

**Table 2 jcm-10-02352-t002:** Screening software results for the DR absent group (numbers represent patients, each with 2 images per eye).

	IDX-DR	Retinalyze Strategy 1	Retinalyze Strategy 2
Marked as DR detected	5	31	7
Marked as no DR	105	79	103
Marked as insufficient quality	0	0	0

DR—diabetic retinopathy.

**Table 3 jcm-10-02352-t003:** A summary of the screening software accuracy results.

	IDX-DR	Retinalyze Strategy 1	Retinalyze Strategy 2
Sensitivity	93.33%	89.66%	74.14%
Specificity	95.45%	71.82%	93.64%
Positive Predictive Value	91.80%	62.65%	86.00%
Negative Predictive Value	96.33%	92.94%	87.29%
Area under the ROC curve	94.39%	80.74%	83.89%
Youden’s index	88.79%	61.47%	67.77%

## Data Availability

The datasets during and/or analyzed during the current study available from the corresponding author on reasonable request.

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
