# Peer review of "Analysis and Comparison of Two Artificial Intelligence Diabetic Retinopathy Screening Algorithms in a Pilot Study: IDx-DR and Retinalyze"

_jcm, 2021, doi:10.3390/jcm10112352_

Round 1

Reviewer 1 Report

In the revised manuscript the authors incorporated or addressed most of the reviewer's suggestions. There are only minor comments left:

  1. The Methods Section still lacks a paragraph on statistics.
  2.  The authors do not define the term referable DR.
  3. It would still be nice to show a table with characterization of DR severity used in the cohort based on the ICO Guidelines.
  4. line 140 should probably refer to DR absent group (since it refers to Table 2) 

A more detailed Response to Reviewer listing all changes made in the manuscript would have been helpful.

Author Response

We would like to thank the reviewer for his/her valuable comments. We amended the manuscript according to the reviewer’s suggestions and present below point by point response to these comments.

Reviewer 1

In the revised manuscript the authors incorporated or addressed most of the reviewer's suggestions. There are only minor comments left:

  1. The Methods Section still lacks a paragraph on statistics.
    We have added a summary of statistics used.
  2.  The authors do not define the term referable DR.
    This was amended according to the reviewer’s suggestion.
  3. It would still be nice to show a table with characterization of DR severity used in the cohort based on the ICO Guidelines.

Unfortunately, we do not have those statistics prepared. The change to exclude mild DR from the DR positive group, excluded 5 patients with mild DR out of previous total of 65, with most of the remaining 60 having moderate disease.

  1. line 140 should probably refer to DR absent group (since it refers to Table 2) 
    This was amended according to the reviewer’s suggestion.

Best regards,

Andrzej Grzybowski, MD, PhD

Professor of Ophthalmology

Reviewer 2 Report

I would first like to thank the authors for this revised version of the manuscript. I has been significantly improved as a result of the changes. I think that the manuscript is still of interest to the readership of the journal. 

I believe that the manuscript requires additional revisions prior to publication. Many of these are to correct for errors made after revision, I believe.

  • Abstract Line 15: Please change to say that TWO comparison studies exist.
  • Line 84-85: The first two sentences of this paragraph are duplicative of the IDx-DR description in the paragraph starting on line 57 and can be removed.
  • Line 101: The word 'centred' should be changed to ‘centered’
  • Line 105: You mention that you had the single physician review the images for evidence of referable DR. However, you do not define rDR. This should be included. You should also reference the use of the international classification of diabetic retinopathy (ICDR) scale during the physician review.
  • Line 128: Include the word 'of' to read "out of four"
  • Line 130: You include two values for Retinalyze agreement, 88% and 89.66%. I believe this is an error and should be fixed.
  • In line 128, you mention that 2 patients were graded as 'insufficient quality' by Retinalyze, but in Table 1 you say that only 1 patient was graded as 'insufficient quality'. Please correct for accuracy and consistency.
  • Line 140: The title of Table 1 says it gives information about rDR positive patients. I believe this should instead reference 'rDR negative' patients. Please correct.
  • Table 2: The total number of patients in each column should total 120 correct? Why do they add to 110?
  • Line 193: Do not use term ‘in this study’ as it makes it seem you are referencing your own results rather than the Tufail et al study
  • Line 209: Similarly, do not say ‘in this study’
  • Line 276: remove ‘a’ in "required a significantly"
  • Line 286: remove ‘the’ from ‘to the both algorithms’

Thank you

Author Response

We would like to thank the reviewer for his/her valuable comments. We amended the manuscript according to the reviewer’s suggestions and present below point by point response to these comments.

Reviewer 2

I would first like to thank the authors for this revised version of the manuscript. It has been significantly improved as a result of the changes. I think that the manuscript is still of interest to the readership of the journal. 

I believe that the manuscript requires additional revisions prior to publication. Many of these are to correct for errors made after revision, I believe.

  • Abstract Line 15: Please change to say that TWO comparison studies exist.
    This was amended according to the reviewer’s suggestion.
  • Line 84-85: The first two sentences of this paragraph are duplicative of the IDx-DR description in the paragraph starting on line 57 and can be removed.
    This was amended according to the reviewer’s suggestion.
  • Line 101: The word 'centred' should be changed to ‘centered’
    This was amended according to the reviewer’s suggestion.
  • Line 105: You mention that you had the single physician review the images for evidence of referable DR. However, you do not define rDR. This should be included. You should also reference the use of the international classification of diabetic retinopathy (ICDR) scale during the physician review.
    This was amended according to the reviewer’s suggestion.
  • Line 128: Include the word 'of' to read "out of four"
    This was amended according to the reviewer’s suggestion.
  • Line 130: You include two values for Retinalyze agreement, 88% and 89.66%. I believe this is an error and should be fixed.

This was amended according to the reviewer’s suggestion.

  • In line 128, you mention that 2 patients were graded as 'insufficient quality' by Retinalyze, but in Table 1 you say that only 1 patient was graded as 'insufficient quality'. Please correct for accuracy and consistency.
    This was amended according to the reviewer’s suggestion.
  • Line 140: The title of Table 1 says it gives information about rDR positive patients. I believe this should instead reference 'rDR negative' patients. Please correct

The tables have been reorganized from the previous version; current titling appears to be correct after re-checking.

  • Table 2: The total number of patients in each column should total 120 correct? Why do they add to 110?

The total number of DR negative patients after is 110. It appears the total n=120 for the DR negative group was inserted by mistake after merging a previous version. This has been amended

  • Line 193: Do not use term ‘in this study’ as it makes it seem you are referencing your own results rather than the Tufail et al study
    This was amended according to the reviewer’s suggestion.
  • Line 209: Similarly, do not say ‘in this study’
    This was amended according to the reviewer’s suggestion.
  • Line 276: remove ‘a’ in "required a significantly"
    This was amended according to the reviewer’s suggestion.
  • Line 286: remove ‘the’ from ‘to the both algorithms’
    This was amended according to the reviewer’s suggestion.

Best regards,

Andrzej Grzybowski, MD, PhD

Professor of Ophthalmology

This manuscript is a resubmission of an earlier submission. The following is a list of the peer review reports and author responses from that submission.

Round 1

Reviewer 1 Report

In this retrospective pilot study, the authors compared two commercially available, AI-based screening systems for diabetic retinopathy. They compared the sensitivity and specificity for detecting DR between the IDx-DR system using 2 fundus images per eye and the Retinalyze System using 1 (Strategy1 ) or 2 (Strategy 2) Fundus images. The authors find that most of the selected images could be analyzed by both systems. The IDx-DR analysis appears to have a higher sensitivity and specificity than the Retinalyze analysis. Interestingly comparing the two strategy groups within the Retinalyze group, the sensitivity frops while the specificity increases when two fundus-images are analyzed.

The paper is clearly structured and well-written. The authors nicely discuss the limitations of their pilot study. I do, however, have the following suggestions/concerns:

  1. The authors state that they selected a set of 65 DR positive screening subjects and 125 DR negative patient images. Most of those images were evaluable by both AI systems. Since the images were, however, taken by untrained personal (a real life setting) it would be important to state how many images were taken altogether and how many per cent of the images were deemed usable for the analysis. It would be even more helpful to load all pictures taken into the AI-based screening software and compare the amount of images that the systems could evaluate.
  2. Characterizing the DR study population with regards to disease severity according to EDTRS criteria would be important with regards to the question if the different AI systems are stronger in specific sub-group (mild DR vs. proliferative DR).
  3. I don’t really understand the difference between Table 1 and 3. Table 1 seems to only represent sensitivity and specificity for DR group while  Table 3 calculates the values for all patients? An additional Statistics paragraph detailing the calculations is necessary in the Material and Methods sections.
  4. The authors contradict themselves with regards to the ophthalmologist reader being blinded: In the M&M section line 94/95 they state that the reads is blinded, in the discussion line 192 they state that the reader was not blinded. Please clarify.
  5. It would be helpful to have at least two ophthalmologist readers to provide an indication for the variability in the grading based on the selected fundus images between graders.

Minor comment:
1. Methodology 1st paragraph: The information that two fundus images were taken is mentioned twice in this paragraph. Please shorten.

Author Response

We would like to thank the reviewer for his/her valuable comments. We amended the manuscript according to the reviewer’s suggestions and present below point by point response to these comments.

Reviewer 1

In this retrospective pilot study, the authors compared two commercially available, AI-based screening systems for diabetic retinopathy. They compared the sensitivity and specificity for detecting DR between the IDx-DR system using 2 fundus images per eye and the Retinalyze System using 1 (Strategy1 ) or 2 (Strategy 2) Fundus images. The authors find that most of the selected images could be analyzed by both systems. The IDx-DR analysis appears to have a higher sensitivity and specificity than the Retinalyze analysis. Interestingly comparing the two strategy groups within the Retinalyze group, the sensitivity frops while the specificity increases when two fundus-images are analyzed.

The paper is clearly structured and well-written. The authors nicely discuss the limitations of their pilot study. I do, however, have the following suggestions/concerns:

  1. The authors state that they selected a set of 65 DR positive screening subjects and 125 DR negative patient images. Most of those images were evaluable by both AI systems. Since the images were, however, taken by untrained personal (a real life setting) it would be important to state how many images were taken altogether and how many per cent of the images were deemed usable for the analysis. It would be even more helpful to load all pictures taken into the AI-based screening software and compare the amount of images that the systems could evaluate.

Although we agree with the reviewer, unfortunately, due to the constraints regarding number of images we were allowed to analyze we had to extensively narrow the scope of our comparison. A decision was made at the beginning of the study to include only the good quality encounters as we were dealing with a large amount of low quality images at the very beginning of our screening initiative.

  1. Characterizing the DR study population with regards to disease severity according to EDTRS criteria would be important with regards to the question if the different AI systems are stronger in specific sub-group (mild DR vs. proliferative DR).

This is another good point that we hope to include in our future study which we are currently planning, with the hindsight gleaned from this pilot study.

  1. I don’t really understand the difference between Table 1 and 3. Table 1 seems to only represent sensitivity and specificity for DR group while Table 3 calculates the values for all patients? An additional Statistics paragraph detailing the calculations is necessary in the Material and Methods sections.

This was made clearer

  1. The authors contradict themselves with regards to the ophthalmologist reader being blinded: In the M&M section line 94/95 they state that the reads is blinded, in the discussion line 192 they state that the reader was not blinded. Please clarify.

We have clarified that the reader was blinded to AI systems’ output at the time of grading

  1. It would be helpful to have at least two ophthalmologist readers to provide an indication for the variability in the grading based on the selected fundus images between graders.

We agree with the reviewer, however that was unfortunately outside the scope of this pilot study, we are currently underway on a larger study with multiple graders.

Minor comment:
1. Methodology 1st paragraph: The information that two fundus images were taken is mentioned twice in this paragraph. Please shorten.

This was amended according to the reviewer’s suggestion

Best regards,

Andrzej Grzybowski, MD, PhD

Professor of Ophthalmology

Reviewer 2 Report

I want to thank the authors for this interesting comparison of two separate AI-based diabetic retinopathy screening technologies. Indeed, few if any direct comparisons of this nature have been performed in the literature.

I would recommend revision to the following areas of the manuscript.

  • Line 16 in abstract - add the word 'a', to make this say "choosing such a system is not ‘a’ simple matter."
  • Background:
    • Please cite the 2016 Tufail et al paper as the only prior comparison in the introduction. Briefly mention the differences in study design and technologies and mention that this present study is adding to the comparison literature
    • Line 37 - change ‘number’ to ’rate’
    • Line 38 - ‘what’ to ‘which’
    • Line 42 -  remove ‘ any time of day or night’, and instead comment on decreased cost, increased access, and lower human capital needs
    • When you first mention Retinalize, IDx-DR, and Eyeart, please cite these references appropriately. Currently, these are not cited with their location of production or any references
    • In the description of Retinalyze: describe whether AI is used in Retinalyze (not clear) and if it requires human interpretation or oversight. Mentino if it still required 35mm film or works on other modalities. Which cameras/types of images can it work on?
    • May want to mention here or in the Methods (or both) that IDx-DR is meant to detect referrable DR, not just any level of DR. This is meaningful difference between the programs.
    • Line 52 - change to "Initially it 'was' composed of..."
    • Line 55 - remove 'or desired'
    • Line 59-60 - I would recommend removing 'owing to a preregistered clinical trial managed by an independent contract research organization'
    • The sentence starting on line 57 is ling and a run-on sentence. Please restructure for clarity and ease of reading by the audience.
    • Line 64 - Please change 'based on' to 'compared to'
    • Line 67 - change to read "Very recently, 'an additional' DR..."
    • Line 67 - remove the '-'
    • Line 73 - remove 'on'
    • Line 77 - remove the comma after 'visited'
  • Methods:
    • Please highlight that IDx-DR is meant to detect referrable DR, which may miss mild DR. 
    • Mention explicitly what the ophthalmologist was detecting. Any evidence of DR? This would differ from IDx-DR output in my understanding.
  • Results:
    • Table 1 and Table 3 are similar. PLease combine. Also, title of Table 1 is misleading as it includes data from more than just the 'DR present group' in calculating specificity.
    • Table 2 - please put total patient number of 'n=125' somewhere in the title or table to give overall numbers
  • Discussion:
    • The first sentence is confusing. Please remove or reformat please.
    • The first paragraph of the discussion should instead be placed in the Methods section to describe how the systems function.
    • You cite your previous paper (citation 11) but do not put it in context with the rest of the paper. Please do so.
    • Paragraph starting on 146 - please explain research and clinical significance of the differences in the two programs (i.e. what it means for docs to not be able to see the IDx-DR assumptions but to have it highlighted for you with Retinalyze). Does it push you one way or the other in terms of usability?
    • Paragraph 155 - Highlight differences in AI use in the early cases/study early on in the paragraph and also highlight other differences in study design. Also highlight lack of direct comparison studies using modern AI based machines. Please qualify any direct comparisons between your study and the 2016 study as the programs were not the same and the methods'study designs were very different.
    • Line 162 - change ‘substantially’ to ‘substantial’
    • Line 165 - explain what you mean by ‘far greater robustness'
    • Paragraph on Line 175 - in addition to the european classifications, please mention FDA classifications of each device as the readership of this paper will be broad and international.
    • Line 183 - please confirm that Retinalyze is a Class I status instead of assuming
    • At the end of the discussion section, I would suggest you include an overall comparison of the two programs, their respective pros and cons, and perhaps ideal use cases for each
  • Limitations:
    • first sentence of second paragraph should be moved to the very first sentence of this section
    • the second paragraph regarding proportion of DR patients seems to be duplicative of first paragraph. Please combine.
    • You mention that you relied on a single ophthalmologist. Please combine this thought with paragraph starting on line 208.
    • The ophthalmologist graded disease severity based on solely a photo, not based on clinical exam. Please make this clear.
    • Please explain why you chose the 'one' or 'two' images approach for Retinalyze, how you expect that to be different from real world use, and how it limits generalizability and ability to compare systems.
  • Conclusion:
    • I believe the conclusion could use a major revision. As currently drafted, it is poorly stated. Please don’t include data. Instead highlight major takeaways from the paper as a whole.

Author Response

We would like to thank the reviewer for his/her valuable comments. We amended the manuscript according to the reviewer’s suggestions and present below point by point response to these comments.

Reviewer 2

I want to thank the authors for this interesting comparison of two separate AI-based diabetic retinopathy screening technologies. Indeed, few if any direct comparisons of this nature have been performed in the literature.

I would recommend revision to the following areas of the manuscript.

  • Line 16 in abstract - add the word 'a', to make this say "choosing such a system is not ‘a’ simple matter."

It was amended according to reviewer’s comment.

  • Background:
    • Please cite the 2016 Tufail et al paper as the only prior comparison in the introduction. Briefly mention the differences in study design and technologies and mention that this present study is adding to the comparison literature

It was amended according to reviewer’s comment

  • Line 37 - change ‘number’ to ’rate’

It was amended according to reviewer’s comment.

  • Line 38 - ‘what’ to ‘which’

It was amended according to reviewer’s comment.

  • Line 42 -  remove ‘ any time of day or night’, and instead comment on decreased cost, increased access, and lower human capital needs

It was amended according to reviewer’s comment.

  • When you first mention Retinalize, IDx-DR, and Eyeart, please cite these references appropriately. Currently, these are not cited with their location of production or any references

It was amended according to reviewer’s comment.

In the description of Retinalyze: describe whether AI is used in Retinalyze (not clear) and if it requires human interpretation or oversight. Mentino if it still required 35mm film or works on other modalities. Which cameras/types of images can it work on? 

It was amended according to reviewer’s comment.

  • May want to mention here or in the Methods (or both) that IDx-DR is meant to detect referrable DR, not just any level of DR. This is meaningful difference between the programs.

It was amended according to reviewer’s comment.

  • Line 52 - change to "Initially it 'was' composed of..."

It was amended according to reviewer’s comment.

  • Line 55 - remove 'or desired'

It was amended according to reviewer’s comment.

  • Line 59-60 - I would recommend removing 'owing to a preregistered clinical trial managed by an independent contract research organization'

It was amended according to reviewer’s comment.

  • The sentence starting on line 57 is ling and a run-on sentence. Please restructure for clarity and ease of reading by the audience.

It was amended according to reviewer’s comment.

  • Line 64 - Please change 'based on' to 'compared to'

It was amended according to reviewer’s comment.

  • Line 67 - change to read "Very recently, 'an additional' DR..."

It was amended according to reviewer’s comment.

  • Line 67 - remove the '-'

It was amended according to reviewer’s comment.

  • Line 73 - remove 'on'

It was amended according to reviewer’s comment.

  • Line 77 - remove the comma after 'visited'

It was amended according to reviewer’s comment.

  • Methods:
    • Please highlight that IDx-DR is meant to detect referrable DR, which may miss mild DR. 

It was amended according to reviewer’s comment.

  • Mention explicitly what the ophthalmologist was detecting. Any evidence of DR? This would differ from IDx-DR output in my understanding.

It was amended according to reviewer’s comment.

  • Results:
    • Table 1 and Table 3 are similar. Please combine. Also, title of Table 1 is misleading as it includes data from more than just the 'DR present group' in calculating specificity.

It was amended according to reviewer’s comment.

  • Table 2 - please put total patient number of 'n=125' somewhere in the title or table to give overall numbers

It was amended according to reviewer’s comment.

  • Discussion:
    • The first sentence is confusing. Please remove or reformat please.

It was amended according to reviewer’s comment.

  • The first paragraph of the discussion should instead be placed in the Methods section to describe how the systems function.

It was amended according to reviewer’s comment.

  • You cite your previous paper (citation 11) but do not put it in context with the rest of the paper. Please do so.

This was amedned according to the reviewier’s comment.

  • Paragraph starting on 146 - please explain research and clinical significance of the differences in the two programs (i.e. what it means for docs to not be able to see the IDx-DR assumptions but to have it highlighted for you with Retinalyze). Does it push you one way or the other in terms of usability?

It was amended according to reviewer’s comment.

  • Paragraph 155 - Highlight differences in AI use in the early cases/study early on in the paragraph and also highlight other differences in study design. Also highlight lack of direct comparison studies using modern AI based machines. Please qualify any direct comparisons between your study and the 2016 study as the programs were not the same and the methods'study designs were very different.

It was amended according to reviewer’s comment.

  • Line 162 - change ‘substantially’ to ‘substantial’

It was amended according to reviewer’s comment.

  • Line 165 - explain what you mean by ‘far greater robustness'

This was reworded and expanded upon.

  • Paragraph on Line 175 - in addition to the european classifications, please mention FDA classifications of each device as the readership of this paper will be broad and international.

It was amended according to reviewer’s comment.

  • Line 183 - please confirm that Retinalyze is a Class I status instead of assuming

We have explained the situation more clearly.

At the end of the discussion section, I would suggest you include an overall comparison of the two programs, their respective pros and cons, and perhaps ideal use cases for each   

It was amended according to reviewer’s comment.

  • Limitations:
    • first sentence of second paragraph should be moved to the very first sentence of this section

It was amended according to reviewer’s comment.

  • the second paragraph regarding proportion of DR patients seems to be duplicative of first paragraph. Please combine.

It was amended according to reviewer’s comment.

  • You mention that you relied on a single ophthalmologist. Please combine this thought with paragraph starting on line 208.

It was amended according to reviewer’s comment.

  • The ophthalmologist graded disease severity based on solely a photo, not based on clinical exam. Please make this clear.

It was amended according to reviewer’s comment.

  • Please explain why you chose the 'one' or 'two' images approach for Retinalyze, how you expect that to be different from real world use, and how it limits generalizability and ability to compare systems.

This was expanded upon in the limitations section, according to the reviewer’s comment

  • Conclusion:
    • I believe the conclusion could use a major revision. As currently drafted, it is poorly stated. Please don’t include data. Instead highlight major takeaways from the paper as a whole.

We have revised the conclusion section according to the reviewer’s comments.

Best regards,

Andrzej Grzybowski, MD, PhD

Professor of Ophthalmology
